# GAP: Scalable Driving with Generative Aided Planner

## Abstract

The primary challenge in end-to-end autonomous driving lines in how to establish robust environmental perception and representations. While most methods improve these capabilities by introducing auxiliary perception tasks, the process of obtaining precise large-scale annotations in this paradigm is both time-consuming and laborious, thereby limiting the scalability and practical application. To address this, we propose an architecture based on the Generative Aided Planner (GAP), which integrates scene generation and planning within a single framework. To compensate for the information loss in discrete image features, we design a dual-branch image encoder that fuses continuous and discrete features, improving the model's ability to recognize traffic lights. Through the scene generation task from input tokens, our approach learns the intrinsic dependencies between tokens and environments, which in turn benefits the planning task. It is important to note that the generative model is trained in a fully self-supervised manner, requiring no perception annotations. Our model is built upon GPT-2, which exhibits scaling laws similar to those observed in other GPTs: as we increase the model size and data size, the performance shows continuous and non-saturating improvements. Experiments show that among methods using the front view as input, our approach outperforms other methods that employ multiple perception supervision in the CARLA simulator. Our method is simple yet highly effective, offering a promising direction for scalable and practical deployment of autonomous vehicles in real-world settings.

## 1 Introduction

End-to-end autonomous driving demonstrates clear advantages over traditional modular approaches by simplifying system architecture, reducing error accumulation and enabling global optimization (Chen et al., 2024). The vanilla end-to-end driving planners often rely on directly optimizing the planning module without intermediate steps (Codevilla et al., 2019; Chen et al., 2020), which focuses solely on the planner's performance, as illustrated in Figure 1(a). Due to the sparse supervision, this paradigm struggles to learn effective environmental representations, resulting in unexpected driving performance. Instead, multi-task-assisted planning improves environmental representations by introducing auxiliary supervised signals, depicted in Figure 1(b). This scheme leverages multiple perception tasks as intermediate steps for achieving supervised learning, such as detection (Chen & Krähenbühl, 2022), map construction(Jiang et al., 2023), and motion prediction(Hu et al., 2023b; Shao et al., 2023), to facilitate the comprehensive understanding of the environment. However, obtaining precise annotations of these perception tasks is time-consuming and laborious at a large scale (Sun et al., 2020; Caesar et al., 2020), thus limiting its scalability and practical application.

To address this limitation, we present an integrated generation and planning framework, termed GAP. As shown in Figure 1(c), our approach utilizes a fully self-supervised generative model to learn representations of the environment, which in turn facilitates more accurate and reliable planning. Our method draws inspiration from recent advancements in large language models (LLMs), such as GPT-series (Radford et al., 2018; 2019; Brown et al., 2020), which have demonstrated the effectiveness of self-supervised learning in sequences modeling by capturing complex patterns and dependencies. Similarly, our framework leverages the power of GPT-like sequences modeling to learn meaningful representations of driving environments without the need of explicit perception supervision. This self-supervised paradigm enables continuous learning from data collected by mil-

Figure 1: (a) Vanilla Approach: A direct optimization of the planning module without intermediate steps, focusing solely on the planner's performance. (b) Multi-Task-Assisted Planning: An approach that leverages multiple perception tasks, including detection, map construction, motion prediction, and others, to enhance the planning capabilities. The accurate perception annotations are often *costly and difficult to acquire at a large scale*. (c) Our integrated generation and planning framework, which utilizes a fully self-supervised generative model to learn representations of the environment, in turn facilitating the model to produce more accurate and reliable planning outcomes.

lions of vehicles at a very low cost (also referred to as fleet learning (Wayve, 2024)), thus benefiting from diverse driving scenarios and experiences of human drivers.

Our "generative aided" approach is primarily implemented based on the GPT autoregressive architecture. Due to the significant gap between the planning and generation tasks, several challenges arise. Firstly, there is a difference in output forms: generation produces dense, pixel-level outputs, whereas planning outputs are sparse. To address this, we take the action query embedding as the autoregressive network's input, while proposing the joint optimization of action regression loss and autoregressive classification loss. Secondly, the generation task focuses more on image details, whereas the planning task commonly relies more on the high-level semantic information and global knowledge. To supplement the information required for planning, we incorporate driving-oriented feature inputs. We adopted a unified architecture that simultaneously outputs planning and generation, with most parameters shared between the two tasks, differing only in their output heads. Thus, the optimization of the generation task can directly influence planning.

We are surprised to find that our approach exhibits properties similar to GPT's scaling laws, showing a consistent and non-saturating performance improvements with increased model size or data size. With the configuration of 345 million parameters (GPT-2 medium) and 256 hours of driving data, our camera-only method achieves a new state-of-the-art (SOTA) in the CARLA simulator (Dosovitskiy et al., 2017), even surpassing those methods that require additional perception supervision.

In summary, our contributions include:

- We propose a novel generative aided planning framework with elaborate model designs, requiring no post-processing and perceptual supervision, and have achieved a new state-of-the-art on the CARLA simulator.

- An empirical examination validates the scaling laws of the proposed model, initially mirroring the appealing properties of large language models.

- We will open-source the code and models to foster the development of the end-to-end autonomous driving community.

## 2 RELATED WORKS

In this section, we discuss end-to-end autonomous driving methods, the applications of generative models in autonomous driving and the scaling laws.

### 2.1 END-TO-END AUTONOMOUS DRIVING

End-to-end autonomous driving has emerged as a hot research topic, replacing traditional rule-based and modular-based approaches by directly learning driving policy from driving videos. Since end-to-end autonomous driving was first introduced over 30 years ago (Pomerleau, 1988), many methods

have been introduced, which can be broadly categorized into two types: reinforcement learning (RL) based methods and imitation learning (IL) based methods. The topic of this work falls under the category of imitation learning. IL learns driving policy directly from expert demonstration data without the need for trial and error exploration of the environment, resulting in higher data efficiency compared with RL based methods. CILRS (Liang et al., 2018) uses a ResNet perception module to process an input image into a latent space, followed by two control prediction heads, which does not utilize temporal information or auxiliary supervision signals, resulting in relatively low performance. Roach (Liang et al., 2018) introduces a stronger RL-based expert model that translates perception ground truth into a Bird's Eye View (BEV) for action prediction. Roach's student model utilizes supervision of both action and intermediate features from the expert model. Following methods such as TCP (Wu et al., 2022) adopts this strategy, employing intermediate features for supervision. These approaches indirectly utilizes perception ground truth as supervision. TCP proposes a rule-based fusion scheme of trajectory and control, improving driving performance across multiple scenarios, but this method requires careful tuning of hyperparameters. Interfuser (Shao et al., 2022) and Transfuser (Chitta et al., 2022) design a post-processing strategy that fuses planning and detection by constraining the planning trajectory to avoid overlapping with detected objects. This approach is limited by the accuracy of perception and does not allow for fully end-to-end optimization. To achieve a more accurate understanding of the environment, most methods utilize specific perception auxiliary tasks for supervision. High-definition maps are used by most methods (Hu et al., 2022; Shao et al., 2022; 2023; Chitta et al., 2022; Chen & Krähenbühl, 2022; Jia et al., 2023b;a) because they provide information on traffic lights, road signs, and the complex topology of road intersections. Additionally, many method (Hu et al., 2022; Shao et al., 2022; 2023; Chitta et al., 2022; Chen & Krähenbühl, 2022; Jia et al., 2023b;a) utilize information on obstacle positions, either through detection boxes or BEV segmentation. Although these multi-task perception aided methods contribute to learning better environmental representations, they also constrain large-scale real-world deployment due to the cost of annotation.

Recently, CarLLaVA (Renz et al., 2024) demonstrated promising performance in autonomous driving using only camera inputs without perception labels. It builds on LLaVA-NeXT's vision encoder pre-trained on internet-scale vision-language data and employs a semi-disentangled output representation combining path predictions and waypoints. While both CarLLaVA and our approach reduce label dependency, we focus on learning representations through generative modeling rather than vision-language pre-training. Our method also exhibits clear scaling properties with increased model and dataset size.

## 2.2 Generative Models for Autonomous Driving

VideoGPT (Yan et al., 2021) leverages a simple GPT-like architecture to autoregressively generate discrete latents. Despite its simplicity, it shows comparable performance with GANs for video generation. GAIA-1 (Hu et al., 2023a) continues this autoregressive paradigm to generate future videos and, with larger model and datasets, exhibits surprising emerging properties. MUVO (Bogdoll et al., 2023) proposes a multimodal generative world model by utilizing raw camera and lidar data to learn a sensor-agnostic geometric representation of the world. Some methods (Zhao et al., 2024; Lu et al., 2024; Zhang et al., 2024; Yang et al., 2024) use diffusion models to generate more realistic future videos. These methods primarily focus on improving the realism of generated videos to be used as neural simulators and are not well-suited for directly planning output. Other methods like Wang et al. (2023b;a) combine ego vehicle trajectory prediction with future video generation, while Zheng et al. (2023) proposes a GTP-style architecture for 4D occupancy prediction. However, their planning effectiveness remains unverified. Importantly, these approaches lack navigation input and rely solely on historical trajectories for extrapolation, leading to issues of causal confusion, as highlighted in Zhai et al. (2023). Among all the methods we have studied, the one closest to ours is MILE (Hu et al., 2022). MILE can simultaneously perform end-to-end planning, BEV segmentation, and RGB image reconstruction. MILE adopts a VAE-like structure to compress observations into a global latent representation, and shows that input images can be decoded from the latent space. However, there is a significant information bottleneck in the model design, preventing the driving task from benefiting from the image generation task. Compared to MILE, our method does not rely on BEV segmentation supervision and demonstrates the benefits of generative tasks for planning through a scalable model architecture.

## 2.3 SCALING LAWS

Scaling laws (Henighan et al., 2020; Kaplan et al., 2020) illustrate mathematical relationships demonstrating how the model performance improves based on various factors such as model size, dataset size, and computing resources. Scaling laws offer numerous benefits. For example, they serve as a guiding principle in model design, facilitating the selection of an optimal scale that balances performance and computational costs. Moreover, scaling laws helps researchers identify important factors that affect improving model performance. Language models like GPT-series (Radford et al., 2018; 2019; Brown et al., 2020; Ouyang et al., 2022), exemplify the principles of scaling laws. As their size increases, measured by parameters or training data, their performance improves. This shows a continuous and non-saturating improvement in performance, validating their long-term advantage in enhancing model capability. Despite the significant success of scaling laws in language models, to our knowledge, there is currently no public paper to study the application of scaling laws in end-to-end autonomous driving. Through examples from language models, we can see the potential of scaling laws in enhancing the performance of end-to-end autonomous driving.

## 3 METHOD

In this section, we delineate the design details of GAP, and the overall architecture is illustrated in Figure 2. In the Section 3.1, we formulate the autoregressive modeling and image tokenization. A detailed explanation of how all inputs are encoded into embeddings is provided in Section 3.2. We introduce the concept of "generative aided" and its implementation in Section 3.3.

### 3.1 PRELIMINARY

**Autoregressive Modeling.** Given a sequence of discrete tokens $\boldsymbol{x} = (x_1, x_2, \ldots, x_n)$, where each token $x_i \in [K]$ is an integer from a vocabulary of size $K$. In an autoregressive model, the probability of token $x_i$ depends on the sequence of preceding tokens $(x_1, x_2, \ldots, x_{i-1})$. According to Bayes' law, we can factorize the likelihood of the sequence $\boldsymbol{x}$ into a product of $n$ conditional probabilities:

$$p(x_1, x_2, \ldots, x_n) = \prod_{i=1}^{n} p(x_i \mid x_1, x_2, \ldots, x_{i-1}). \tag{1}$$

This factorization is also known as *next-token prediction*. The training process is purely self-supervised and typically aims to maximize the joint probability of the sequence. During training, the teacher forcing method is employed, which allows the next token to be predicted using fully parallel computation.

**Tokenization.** The tokenization of images serves two main purposes. Firstly, the encoder compresses information by mapping RGB pixel values to a more compact representation space, thereby improving the computational efficiency of subsequent generative models. Secondly, the quantizer discretizes continuous representations by mapping them to *discrete* integer values, which is consistent with the next-token prediction of autoregressive models. This can be described by the following equations:

$$f = \mathcal{E}(\boldsymbol{I}), \qquad x = \mathcal{Q}(f), \tag{2}$$

where $\boldsymbol{I}$ is the input image, $\mathcal{E}(\cdot)$ an encoder, and $\mathcal{Q}(\cdot)$ a quantizer. Typical works such as VQ-VAE (Van Den Oord et al., 2017) and VQGAN (Esser et al., 2020) involve a learnable codebook $\boldsymbol{e} \in \mathbb{R}^{K \times C}$ containing $K$ vectors. The quantization function $x = \mathcal{Q}(f)$ maps each feature vector $f^{(i,j)}$ to the index $x^{(i,j)} \in [K]$ of its nearest codebook embedding:

$$x^{(i,j)} = \underset{k \in [K]}{\arg\min} \|\boldsymbol{e}(k) - f^{(i,j)}\|_2, \tag{3}$$

where $\boldsymbol{e}(k)$ means the $k$-th vector of the codebook $\boldsymbol{e}$. During training, the decoder $\mathcal{D}(\cdot)$ reconstruct the image, and the difference can be messured by $\epsilon$, that is:

$$\hat{\boldsymbol{I}} = \mathcal{D}(\boldsymbol{e}(x)), \qquad \epsilon = \|\boldsymbol{I} - \hat{\boldsymbol{I}}\|_2. \tag{4}$$

Due to the presence of $\epsilon$, some detailed information may be lost, *significantly* impacting the performance of the method. Our solution to this issue is provided in Section 3.2.

Figure 2: The core structure of GAP is an autoregressive transformer decoder (GPT-2) on sequences of tokens, incorporating inputs from multiple consecutive timesteps. The speed and navigation are encoded to serve as prompts for driving decisions. The input image is fed into two branches: VQ-GAN converts the image into discrete tokens, while the other extracts driving-oriented information through a lightweight ConvNet to compensate for the information loss caused by vector quantization. The VQGAN branch is fixed during the training process, while the ConvNet branch is trained jointly with GPT-2 model. The token <c> marks the starting flag of image tokens, and it will predict the first token "1" in the image, with the input of token "1" predicting token "2", and so on. The token <a> is the action query, which will be updated by GPT-2 and used to regress the control signals of accelerator, braking, and steering.

## 3.2 TOKEN EMBEDDINGS

The GAP compresses image patches into discrete tokens with VQGAN (Esser et al., 2020):

$$\boldsymbol{x}_t = \mathcal{F}_q(\boldsymbol{I}_t) \in \mathcal{R}^n, \tag{5}$$

where $\boldsymbol{I}_t$ is the input RGB image at time $t$, the model $\mathcal{F}_q = \mathcal{Q}(\mathcal{E}(\cdot))$, with a spatial downsampling rate of 16, with $n = h \times w$, and the vocabulary embedding space $\boldsymbol{e} \in \mathcal{R}^{K \times C}$, where $K$ represents the number of visual words and $C$ denotes the dimensionality of each word embedding. $K$ is 1024 in our model, thereby the bit compression ratio is given by $\frac{16 \times 16 \times 3 \times 8}{\log_2 1024} = 614$.

Considering that vector quantization may result in the loss of fine-grained information, especially for small objects such as traffic lights, we found that only using quantized features will result in a very low route completion rate due to the invisibility of traffic lights (refer to experiments 4.4). To alleviate this issue, we introduce an additional non-quantized branch to extract driving-oriented information. Specifically, we employ a lightweight convolutional network $\mathcal{F}_c$ to obtain image features at 1/64 spatial resolution of the original image, which are then flattened and used as prompts for planning, that is,

$$\boldsymbol{c}_t = \mathcal{F}_c(\boldsymbol{I}_t) \in \mathcal{R}^{(h_c \times w_c) \times C}. \tag{6}$$

For navigation inputs, following the prior works (Hu et al., 2022; Zhang et al., 2021), we first transform the navigation route at time $t$ into a binary mask $\boldsymbol{M}_t$, which is then mapped into a high-dimensional vector by a ResNet-18 (He et al., 2016) network, as:

$$\boldsymbol{r}_t = \mathcal{F}_r(\boldsymbol{M}_t) \in \mathcal{R}^{1 \times C}. \tag{7}$$

Furthermore, we employ $\mathcal{F}_s$, a Multi-Layer Perceptron (MLP), to encode the scalar speed $v_t$ of the ego vehicle as:

$$\boldsymbol{v}_t = \mathcal{F}_s(v_t) \in \mathcal{R}^{1 \times C}. \tag{8}$$

These token embeddings are combined with learnable positional embeddings by default. All tokens are sequentially fed into the autoregressive model in the following order:

$$(\boldsymbol{v}_t, \boldsymbol{r}_t, \boldsymbol{c}_t, \texttt{<c>}, \boldsymbol{e}(\boldsymbol{x}_t), \texttt{<a>}), \ t \in 1, \ldots, T, \tag{9}$$

where $T$ is the number of temporal frames, $\texttt{<c>}$ denotes the starting flag of image tokens, $\texttt{<a>}$ represents the query token for the action of ego vehicle, and both are implemented using learnable embeddings.

### 3.3 GENERATIVE AIDED PLANNER

The concept of "generative aided" can be realized through various methods, such as VAEs, diffusion models, autoregressive models, and so on. However, considering the flexibility of autoregressive models in taking diverse prompts and the remarkable scalability of the GPT series, we have chosen GPT-2 for our generative model. As shown in Figure 2, the model is mainly based on the GPT architecture and is trained with two tasks: autoregressive generation of the next token and action regression for the planner. The probability of the next token is given by $\mathcal{P}_{\theta,\theta_g}(\cdot)$, whereas the prediction of the action is represented by $\mathcal{G}_{\theta,\theta_a}(\cdot)$. It is noteworthy the two tasks share the majority of the parameters ($\theta$), except for the different output heads ($\theta_g$ and $\theta_a$). When optimizing the shared parameters $\theta$ for the generation task, the model will well learn the dependencies of input tokens. This results in a robust representation of the environment, which subsequently benefits the planning task.

**Connection with world models.** World models LeCun (2022) similarly learn and understand internal representations of the environment through self-supervised tasks. World models typically use ground truth actions as conditions, emphasizing future prediction or environment simulation. However, our method focuses on improving the planning task.

**Training objective.** We reformulate the Equation 1 by incorporating three prompts $(\boldsymbol{v}, \boldsymbol{r}, \boldsymbol{c})$ that described in Section 3.2 and adopt the log-likelihood autoregressive loss function,

$$\mathcal{L}_{\text{gen}} = -\sum_{t=1}^{T} \sum_{i=1}^{n} \log \mathcal{P}_{\theta,\theta_g}(x_{t,i} \mid \boldsymbol{v}_{\leq t}, \boldsymbol{r}_{\leq t}, \boldsymbol{c}_{\leq t}, \boldsymbol{x}_{<t}, \boldsymbol{x}_{t,j<i}), \tag{10}$$

All observations prior to timestep $t$ are used to predict the action and we calculate the $L_1$ loss with the ground truth $\boldsymbol{a}_t$:

$$\mathcal{L}_{\text{action}} = \sum_{t=1}^{T} \|\mathcal{G}_{\theta,\theta_a}(\boldsymbol{v}_{\leq t}, \boldsymbol{r}_{\leq t}, \boldsymbol{c}_{\leq t}, \boldsymbol{x}_{\leq t}) - \boldsymbol{a_t}\|_1. \tag{11}$$

Therefore, the total loss is the sum of the above two loss, weighted by the hyperparameter $\alpha$:

$$\mathcal{L} = \mathcal{L}_{\text{action}} + \alpha \mathcal{L}_{\text{gen}}. \tag{12}$$

## 4 EXPERIMENTS

### 4.1 EXPERIMENTS SETUP

**Datasets.** We utilize CARLA as the simulator for both data collection and closed-loop evaluation. The test routes are chosen from the ten longest routes in Town05, known as Town05Long. We employ two experimental setups for evaluation: The first setup is consistent with TCP (Wu et al., 2022), where both training and testing include challenging driving scenarios. We collect 27 hours of training data for this setup. The second setup aligns with MILE (Hu et al., 2022), for which we gather 32 hours of training data. To ensure temporal compactness of the training data, we collect data at 10 Hz. Furthermore, to validate our model's scalability, we collect data across all 8 towns under 21 weather conditions, gathering approximately 256 hours of driving data, totaling 9.6M frames.

**Training.** Our model is trained for 80k iterations on a total batch size of 64 on 8 A800 GPUs, with training sequence length $T = 6$. The weight decay is 1e-3 for the decoder parameters and 1e-4 for the other parameters. $\alpha$ drops from 1.0 to 0.2 over 50k iterations linearly.

| Method | Postprocess | Modality | Extra Labels | Hours | DS↑ | RC↑ | IS↑ |
|--------|-------------|----------|--------------|-------|-----|-----|-----|
| Interfuser Shao et al. (2022) | PID+Constraint | C3L1 | Map+Box | 410 | 68.3±1.9 | 95.0±2.9 | - |
| ReasonNet Shao et al. (2023) | PID+Constraint | C4L1 | Map+Box | 55 | 73.2±1.9 | 95.9±2.3 | 0.76±0.03 |
| Transfuser Chitta et al. (2022) | PID | C3L1 | Depth+Seg+Map+Box | 27 | 31.0±3.6 | 47.5±5.3 | 0.77±0.04 |
| LAV Chen & Krähenbühl (2022) | PID | C4L1 | Expert+Seg+Map+Box | 27 | 46.5±2.3 | 69.8±2.3 | 0.73±0.02 |
| ThinkTwice Jia et al. (2023b) | PID+Fusion | C4L1 | Expert+Depth+Seg+Map | 275 | 70.9±3.4 | 95.5±2.6 | 0.75±0.05 |
| DriveAdapter Jia et al. (2023a) | PID+Fusion | C4L1 | Expert+Depth+Seg+Map | 275 | 71.9±0.0 | 97.3±0.0 | 0.74±0.00 |
| CILRS Codevilla et al. (2019) | PID | C1 | None | 27 | 7.8±0.3 | 10.3±0.0 | 0.75±0.05 |
| LBC Chen et al. (2020) | PID | C3 | None | 27 | 12.3±2.0 | 31.9±2.2 | 0.66±0.02 |
| Roach Zhang et al. (2021) | None | C1 | Expert | 27 | 41.6±1.8 | 96.4±2.1 | 0.43±0.03 |
| TCP Wu et al. (2022) | PID+Fusion | C1 | Expert | 27 | 57.2±1.5 | 80.4±1.5 | 0.73±0.02 |
| Ours (GPT2-small) | None | C1 | None | 27 | 57.1±4.0 | 100.0±0.0 | 0.57±0.04 |
| MILE† Hu et al. (2022) | None | C1 | Map+Box | 32 | 61.1±3.2 | 97.4±0.8 | 0.63±0.03 |
| MILE† Hu et al. (2022) | None | C1 | None | 32 | 55.0±3.3 | 92.5±2.4 | 0.61±0.04 |
| Ours†(GPT2-small) | None | C1 | None | 32 | 58.5±1.7 | 96.0±1.3 | 0.60±0.01 |
| Ours†(GPT2-small) | None | C1 | None | 256 | 73.2±1.9 | 93.9±4.1 | 0.78±0.03 |
| Ours†(GPT2-medium) | None | C1 | None | 256 | 77.8±2.6 | 98.1±1.5 | 0.79±0.02 |

Table 1: **Performance on Town05 Long benchmark.** † denotes no specific scenarios are involved in the training and evaluation. Different methods have various configurations; most of our experiments follow the same configuration as MILE (Hu et al., 2022). *Extra labels* refers to perception labels required to train the model besides actions. *Hours* represents the duration of the training dataset. *CxLy* means using x cameras and y LiDARs. *Expert* denotes the distillation from privileged agents' features, which are extracted from multiple kinds of perception labels. *Map* denotes the high-definition map. *Depth* and *Seg* denotes the depth and semantic segmentation labels of the 2D images. *Box* denotes the bounding boxes of surrounding agents.

**Metrics.** We employ the official evaluation metrics from the CARLA leaderboard: Route Completion (RC), which represents the percentage of the route successfully completed by the autonomous driving agent. Infraction Score (IS) measures the count of infractions along the route, including violations involving pedestrians, vehicles, road layout, red lights, and other factors. The primary metric, Driving Score (DS), is the product of Route Completion and the Infraction Score.

## 4.2 COMPARISON WITH OTHER METHODS

In Table 1, we compare our method with previous approaches. Unlike other methods, our approach does not require any additional labels for supervision. Among methods that use only front-view images as input, we achieve driving performance comparable to MILE (Hu et al., 2022) and TCP (Wu et al., 2022) when trained on the same amount of data. While neither method uses perception annotations, our approach outperforms MILE, improving the route completion score from 92.5 to 96.0 without increasing traffic violations. This improvement can be attributed to our generative auxiliary task, which enhances environmental representation learning.

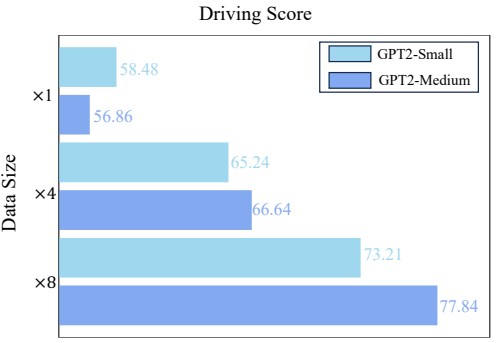

Figure 3: The empirical study of scaling up. We report the Driving Score on Town05Long for different dataset and model sizes. The result is the mean of three runs.

Notably, our method demonstrates significant performance gains when leveraging larger amounts of driving data. When trained on 256 hours of data, our approach achieves driving performance far superior to that of MILE (Hu et al., 2022). This scalability highlights the potential of our method in leveraging large-scale real-world driving datasets.

## 4.3 SCALING UP

Figure 3 demonstrates our approach's scalability in terms of dataset and model size. Using 32 hours of driving data as a baseline: Firstly, with GPT-2 small, increasing the amount of data to 4× and 8× increases the driving score from 58.5 to 65.2 and 73.2 respectively. This demonstrates our method's ability to effectively leverage large unannotated datasets to facilitate the transfer from

simulated to real-world environments. Secondly, larger models benefit from increased data. While parameter increase does not improve performance with baseline data, significant gains are observed with 4× and 8× data, highlighting the synergy between dataset size and model capacity. Thirdly, our experiments provide guidance for choosing parameter and dataset size, so that we can predict performance by interpolation or extrapolation. We have not yet seen any saturation in performance, suggesting there is still room for improvement.

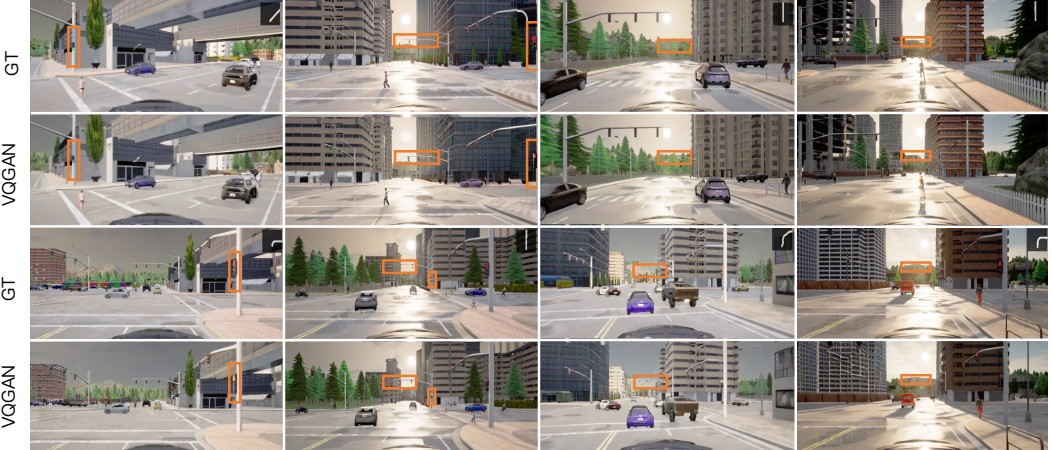

Figure 4: Failed reconstructions of traffic lights by VQGAN. Traffic lights in the scenes are highlighted with orange boxes.

## 4.4 ABLATION STUDIES

**Driving-oriented feature.** Unifying image autoregression and driving tasks poses challenges due to discrete feature limitations. Figure 4 shows VQGAN's poor traffic light reconstruction, illustrating quantization-induced information loss. Table 2 demonstrates that relying solely on discrete features leads to lower route completion rates and more red light violations. To address this, we introduce driving-oriented features. Moreover, the learning process of discrete features is predominantly reconstruction-oriented, which may not align optimally with driving tasks. Table 2 shows that driving-oriented features achieve better overall driving scores despite higher red light violation rates compared to ground truth traffic lights. This suggests that continuous features learned jointly with the driving task capture relevant information missing in discrete features.

| Inputs | Data | Driving Score | Route | Infraction | Red Light |
|---|---|---|---|---|---|
| discrete feature | 32h | 26.90 | 59.04 | 0.58 | 2.25 |
| driving-oriented feature | 32h | 53.23 | 95.01 | 0.55 | 0.46 |
| discrete feature + driving-oriented feature | 32h | 52.52 | 86.13 | 0.64 | 0.34 |
| discrete feature + gt. traffic lights | 32h | 48.68 | 62.21 | 0.83 | 0.05 |

Table 2: Impact of driving-oriented feature and discrete feature. *Red Light* means the count of red light violations per kilometer.

**Next-token prediction.** The next-token prediction task provides a denser and stronger supervision signal than the sparse actions, therefore enhancing the model's understanding of driving scenes. We observe that this auxiliary task improves the detection of dynamic objects in the scene. As shown in Figure 5, by incorporating image generation as an auxiliary task, the model correctly attends to both moving and stationary vehicles. The areas occupied by other vehicles are considered nondrivable, with lower weights during the update of intermediate action features, allowing the model to make correct decisions in the remaining drivable areas. Another interesting observation is the stronger response to the current moment compared to historical ones. This demonstrates the model's ability to handle long-term dependencies effectively, with a greater focus on current

moment. As demonstrated in Table 3, without next-token prediction results in a significant drop in the overall driving score.

| Setting | Driving Score | Route | Infraction |
|---------|---------------|-------|------------|
| w/o next-token prediction | 52.52 | 86.13 | 0.64 |
| w/o random mask | 57.10 | 95.84 | 0.59 |
| Full(Ours) | 58.48 | 96.03 | 0.59 |

Table 3: Ablation studies. We report driving performance on novel towns under new weathers in CARLA. Results are averaged across three runs.

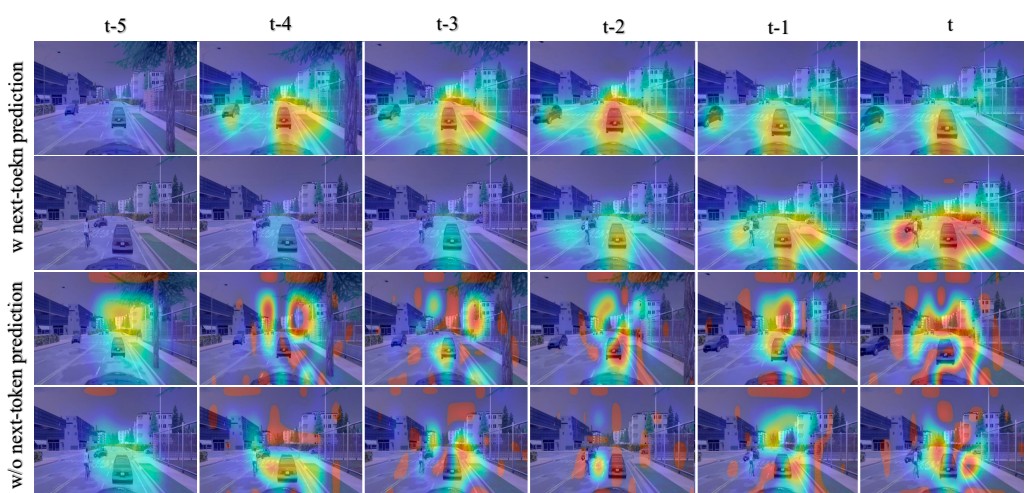

Figure 5: Visualization of attention weights. We select the action token at time $t$ as the query and demonstrate the responses of the driving-oriented feature at different time steps. The blue areas indicate higher attention scores, while the red areas indicate lower attention scores.

**Random mask.**   Images often contain redundant information irrelevant to driving tasks. A simple causal mask for attention may cause the network to focus excessively on local details, neglecting global driving scene information. To address this, we adopt the approach from MAE He et al. (2022), applying large-scale masking to discrete tokens. This technique randomly replaces image tokens with learnable embeddings, compelling the model to learn long-range dependencies.

### 4.5 LIMITATIONS AND IMPACTS

**Limitations.** Previous multi-task-assisted methods utilize the white-box information from perception outputs, which can help debug the reasons for planning errors. Our method does not output perception results, which may reduce its interpretability to some extent.

**Impacts.** Our methodology, which relies solely on human driving data and eliminates the need for supplementary annotations, markedly reduces the cost of autonomous driving. This could potentially expedite the widespread adoption of mass-produced autonomous vehicles.

## 5 CONCLUSION

We propose an end-to-end autonomous driving architecture called the generative aided planner (GAP), which significantly improves the performance of autonomous driving by learning environment representations through self-supervised generative tasks. Built upon GPT-2, our method demonstrates scalability in both data size and model size. Despite its simplicity, our approach yields state-of-the-art results on the CRALA benchmark. In the future, we aim to expand our experiments on scaling laws and adapt this method to a vision-language multimodal framework, offering interpretable textual explanations for autonomous driving decisions.

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

# A APPENDIX

## A.1 DETAILED INFRACTION SCORES OF ABLATION STUDY

| Inputs | Data | Ped. Coll. | Veh. Coll. | Red Light | Stop Infr. | DS | Route | Infraction |
|---|---|---|---|---|---|---|---|---|
| DF | 32h | 0.00 | 0.49 | 2.25 | 0.17 | 26.90 | 59.04 | 0.58 |
| DOF | 32h | 0.14 | 0.33 | 0.46 | 0.98 | 53.23 | 95.01 | 0.55 |
| DF + GT Traffic Light | 32h | 0.03 | 0.30 | 0.34 | 0.85 | 52.52 | 86.13 | 0.64 |
| DF + DOF | 32h | 0.02 | 0.42 | 0.05 | 0.53 | 48.68 | 62.21 | 0.83 |
| DF + GT Traffic Light | 128h | 0.01 | 0.36 | 0.07 | 0.84 | 62.09 | 87.82 | 0.72 |
| DF + DOF | 128h | 0.03 | 0.37 | 0.36 | 0.38 | 65.24 | 95.62 | 0.68 |

Table 4: Impact of driving-oriented features and discrete features. DF: discrete features, DOF: driving-oriented features. Columns 3 to 6 show the count of infractions per kilometer: Ped. Coll. represents pedestrian collisions, Veh. Coll. represents collisions with other vehicles, Red Light indicates running red lights, and Stop Infr. refers to not stopping at stop signs.

| Inputs | Data | Ped. Coll. | Veh. Coll. | Red Light | Stop Infr. | DS | Route | Infraction |
|---|---|---|---|---|---|---|---|---|
| W/o NTP | 32h | 0.03 | 0.30 | 0.34 | 0.85 | 52.52 | 86.13 | 0.64 |
| W/o random mask | 32h | 0.05 | 0.43 | 0.36 | 0.79 | 57.10 | 95.84 | 0.59 |
| Full | 32h | 0.03 | 0.31 | 0.23 | 1.15 | 58.48 | 96.03 | 0.61 |

Table 5: Ablation studies. We denote next-token prediction as NTP.

| sequenth length | Data | Ped. Coll. | Veh. Coll. | Red Light | Stop Infr. | DS | Route | Infraction |
|---|---|---|---|---|---|---|---|---|
| 2 frames | 32h | 0.06 | 0.36 | 0.25 | 1.06 | 52.97 | 95.04 | 0.57 |
| 4 frames | 32h | 0.03 | 0.33 | 0.23 | 0.82 | 62.10 | 93.66 | 0.65 |
| 6 frames | 32h | 0.03 | 0.31 | 0.23 | 1.15 | 58.48 | 96.03 | 0.61 |

Table 6: Impact of different temporal lengths on model performance

## A.2 FAILED CASES

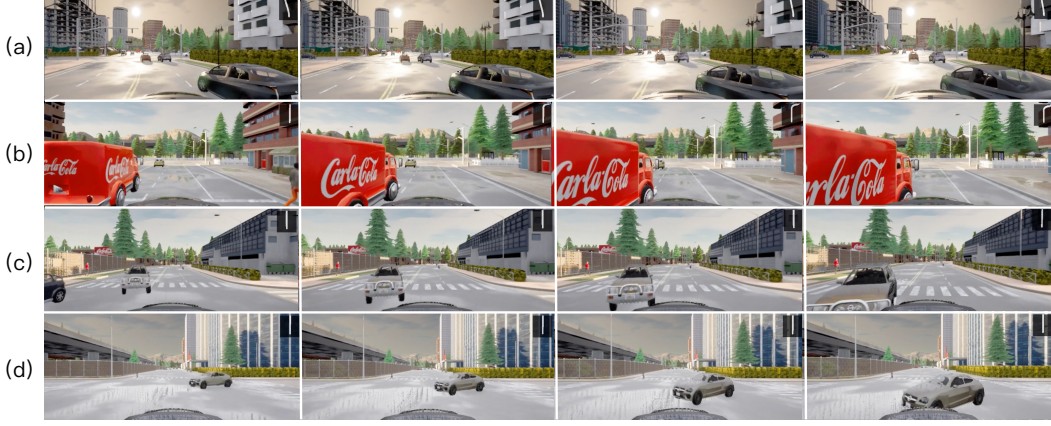

Figure 6: Collision scenarios with other vehicles. (a) and (b) show instances where the ego vehicle failed to maintain a safe distance from vehicles on both sides while moving forward, resulting in an inability to yield in time when those vehicles changed lanes. (c) and (d) depict scenarios at unprotected intersections where the ego vehicle failed to yield in time when interacting with vehicles with unclear intentions.

## A.3 DETAILED MODEL SIZES

| Model | Num Layers | Num Heads | Num embeddings | Param | Inference Time (ms) | |
|---|---|---|---|---|---|---|
| | | | | | Once Forward | Autoregression |
| GPT2-small | 12 | 12 | 768 | 126M | 100 | 3500 |
| GPT2-medium | 24 | 16 | 1024 | 347M | 140 | 4900 |

Table 7: Details of the model parameters for the transformer decoder GPT2-small and GPT2-medium. The inference time is measured on A800 GPU. "Once forward" refers to non-autoregressive prediction, while "Autoregression" indicates autoregressive prediction.

## A.4 QUANTITATIVE RESULTS OF NEXT-TOKEN PREDICTION

| Data | L2(1e-2) | | FID | |
|---|---|---|---|---|
| | Once Forward | Autoregression | Once Forward | Autoregression |
| 32h | 2.70 | 2.70 | 21.24 | 20.70 |
| 128h | 3.27 | 2.68 | 21.59 | 20.20 |
| 256h | 4.23 | 3.25 | 26.49 | 25.19 |

Table 8: Quantitative results of generation quality measured by L2 distance and FID score. Lower is better.

