# OpenReview forum: "GAP: Scalable Driving with Generative Aided Planner"
_ICLR.cc/2025/Conference — Submitted to ICLR 2025_

### Official Review · Reviewer_GDwn · 2024-11-01

**Soundness:** 3
**Presentation:** 4
**Contribution:** 3
**Rating:** 5
**Confidence:** 5

**Summary:**

The paper proposes a full end-to-end driving method, without intermediate supervision,
that uses a tokenization substep (VQGAN) and a language model type architecture (GPT-2).
With this approach, the method demonstrated high quality results on a CARLA closed loop
benchmark.

**Strengths:**

* The main strength of the paper is the proposition of a different architecture for the full end-to-end driving problem.
This proposed architecture is sounding since the tokenization step is likely to reduce potential pixel ambiguity
that exists on pure pixel representation. Then by using a GPT architecture the tokens can be more efficiently used.

* The evaluation, even though limited, shows very compelling driving results. I was particulary happy with the scalability results, which is very important for this type of contributions.

* The driving oriented features are also a solid contribution.

**Weaknesses:**

* Effects of data on the results can be massive. I think the main weakness of the paper is the fact that little is talked about the dataset and different . Over the years, people has been getting better on curating and making good datasets for methods, even the CILRS method has been reported having drastically better results by just increasing input resolution [1] [2]. Changes in the dataset curation can have massive impact. The fact that the paper just mentions that it uses "a dataset of 27 then a dataset of 256  over 8 towns" gives insuficient information. I would like to see how selection of different parts of this dataset really impacts the model. I am not convinced that this method is indeed better than MILE or CIL++ for this application. I would like to be convinced that the tokenization is the way forward but I feel this gives me insufficient information.
Something like training a simple CIL++ or a MILE and GPT2 in the same dataset.  I know this is a lot of work but is necessary since data plays a very crucial role on this application.

* Limited evaluation specially due to tokenization effects on real world datasets. The fact that this paper only does evaluation on CARLA does not convince me that the tokenization approach scale for real world images. Closed loop test is the main . However, CARLA only tests lacks the variability perceptually to validate a tokenization approach of the image space. CARLA scenes are very repetitive and easily tokenizable, even when considering the towns with more assets. There are kind of closed loop
datasets like NuPlan or waymo with the new waymo agents that would allow to test the tokenization approach.

This leads to my third point:

* Lack of comparison with motion prediction  literature.
Even though the use of this architecture is new on the so called End-to-End driving literature it is not new at all on the motion prediction literature with many papers using GPT like architectures like autobots or scene transformers. UNIAD variation papers [3] also incorporate tokenziation even thought they do use extra information than just the pixels. When compared to those methods the novelty of this architecture is a bit more limited.


[1]Hu, Anthony, et al. "Model-based imitation learning for urban driving." Advances in Neural Information Processing Systems 35 (2022): 20703-20716.
[2]Xiao, Yi, et al. "Scaling self-supervised end-to-end driving with multi-view attention learning." arXiv preprint arXiv:2302.03198 2 (2023).
[3] Jiang, B., Chen, S., Xu, Q., Liao, B., Chen, J., Zhou, H., ... & Wang, X. (2023). Vad: Vectorized scene representation for efficient autonomous driving. In Proceedings of the IEEE/CVF International Conference on Computer Vision (pp. 8340-8350).

**Questions:**

I think the weakness section raises enough questions. Please check that specifically.

I miss ablation results for the tokens, What do the token represents and what is the best number there ?

---

> ### Comment · Reviewer_GDwn · 2024-12-02
>
> My concerns stay unfortunately.
>
> I like the method but this is a data domain, data matters more than everything else. The comparison of a solid baseline that uses the same data is fundamental in my opinion. The use of non-simulation pixels also.
>
> I am ok if it gets  accepted but I should stay marginally below acceptance.

---

### Official Review · Reviewer_AoXu · 2024-11-03

**Soundness:** 3
**Presentation:** 3
**Contribution:** 3
**Rating:** 6
**Confidence:** 5

**Summary:**

By proposing the Generative Aided Planner (GAP) that integrates scene generation and planning in a single framework, the paper effectively circumvents the need for costly perception annotations through self-supervised learning. Notably, the dual-branch image encoder and GPT-2-based architecture demonstrate consistent, scalable performance improvements. The approach shows competitive results in the CARLA simulator.

**Strengths:**

1. The manuscript is well-written and easy to follow.
2. The proposed paradigm is innovative and provides a fresh perspective on integrating generative tasks into autonomous driving.
3. The method demonstrates competitive performance, with noticeable improvements as the model scales.

**Weaknesses:**

1. The paper would benefit from including a comprehensive report on latency to better evaluate real-time feasibility.
2. In Fig.3, it is more informative to compare the scaling effects against other existing methods, rather than just contrasting GPT-small and GPT-medium.
3. Including standard metrics such as FID to evaluate generation quality would be valuable for understanding the overall quality of the generated images.
4. Directly outputting control signals has certain limitations, as control signals can vary significantly between different vehicles, making generalization difficult. It is encouraged to report the model's results when outputting waypoints.
5. Based on Figure 5, an investigation into the impact of temporal sequence length on the performance of the autoregressive model could provide valuable insights.
6. The organization of Table 2 is somewhat strange. It would be clearer to present results for discrete features, discrete features + ground truth traffic lights, and discrete features with driving-oriented features under *consistent* data conditions.

**Questions:**

See weakness. I will raise my score if the concerns are well-addressed.

---

### Official Review · Reviewer_ka22 · 2024-11-03

**Soundness:** 2
**Presentation:** 2
**Contribution:** 2
**Rating:** 5
**Confidence:** 5

**Summary:**

In this study, an end-to-end driving approach named GAP is introduced. GAP is constructed using an autoregressive language model, GPT-2, which processes encoded speed, navigational signals, and images to forecast both the image and subsequent action. Notably, the input image is split into two branches: one based on ConvNet to retain environmental intricacies and another based on VQGAN for image reconstruction, serving as an additional task for predicting the next token. GAP is supervised with action and image reconstruction losses and does not require annotations of perception tasks. GAP is assessed using the Town05 Long benchmark within the CARLA simulator. It demonstrates state-of-the-art performance and exhibits scalability concerning both data and model dimensions.

**Strengths:**

1. This paper validates the scaling up phenomenon with the proposed architecture, in two different model sizes and three levels of data size.
1. The proposed method does not require other annotations such as perception or prediction labels, which benefits scaling up with more raw data in the future.
1. The paper and method are easy to follow and understand.

**Weaknesses:**

I think this paper is not presented soundly and rigorously. Some parts are technically incorrect leading to overclaimed contributions.
1. Current usage of GAP is not autoregressive. The model does not predict the next token based on previous predictions. Additionally, if temporal information is utilized is unclear.
2. The model is not fully self-supervised, because it still needs actions for supervision. If this method can be regarded as self-supervision, then a lot of previous works like Transfuser (the original CVPR 2021 version), CILRS, and more, are all self-supervised. This is absolutely unreasonable. Besides, the authors claim that using 'Expert' is also a kind of extra label. However, MILE uses Roach expert, and this paper follows MILE for data collection. That indicates this paper relies on privileged information for expert-based data collection as well. Another thing is, LBC does not use expert feature distillation and Roach either. These incorrect technique details weaken the claimed contributions.
3. Some descriptions when introducing related works or highlighting the differences of this paper are not accurate or sound. For example, Zheng et al. (2023) Line 148 is not a video-based world model. It predicts occupancy. The OccWorld work also proposes a GPT-based model and validates the scaling law in occupancy prediction. Besides, the authors missed an important related work: CarLLaVA [1]. CarLLaVa also proposed a language model-based transformer to scale up the model. The authors should provide more context and comparison regarding these issues clearly or rigorously. Considering this, I think the novelty of this paper needs further discussion.
4. One important weakness of this paper is GAP's most essential factor is not the next token prediction or other claimed contributions. As shown in Table 2, the driving-oriented feature from CNN improves the performance by 25.6 DS, while the next-token prediction enhances only 4.6 DS. The CNN encoding is not novel and is commonly used in each end-to-end driving paper. The most important driving information comes from the ConvNet, instead of the VQGAN network.
5. I do not fully get the motivation of generative modeling. The predictive feature (next-token prediction) is not fully utilized in the framework, based on the above discussions. How about predicting the future and then decoding the action, or using the VQGAN training as a pre-training stage similar to some occupancy prediction world models, similar to methods in robotics [2]?

This paper does not include the leaderboard evaluation results. That would be considered a much more convincing benchmark.

The writing is relatively casual. The format is incorrect for all citations. The authors should use \citep in most cases. There are limited citations in the introduction to support various claims. I cannot grasp the idea in Fig. 1(c) when comparing it with (a-b). Line 268: Multi-layer Perceptron.

[1] Renz, Katrin, et al. "CarLLaVA: Vision language models for camera-only closed-loop driving." arXiv preprint arXiv:2406.10165 (2024).
[2] Yang, Mengjiao, et al. "Learning interactive real-world simulators." arXiv preprint arXiv:2310.06114 (2023).

**Questions:**

Please see the weaknesses part.

---

### Official Review · Reviewer_PHnB · 2024-11-03

**Soundness:** 3
**Presentation:** 2
**Contribution:** 3
**Rating:** 5
**Confidence:** 4

**Summary:**

This paper presents an architecture, Generative Aided Planner (GAP), which integrates scene generation and planning to help the model produce better actions. Built upon GPT-2, GAP could be trained in a self-supervised manner and can be scaled up without the requirements of large-scale annotations. Moreover, a dual-branch structure is designed to use a trainable image encoder to capture features lost by VQGAN. The method is evaluated in CARLA and achieves competitive results.

**Strengths:**

1. Clear illustration of the model structure (Fig. 2), and abundant visualizations.
2. The scaling experiments in Sec. 4.3 are intuitive and valuable, since scaling properties of autonomous driving models are previously underexplored.
3. Outstanding performance of the proposed method on challenging Carla benchmark.

**Weaknesses:**

1. Lack of ablation studies on the importance of VQVAE features. The visual inputs of the proposed methods come from both ConvNet and VQVAE features, but the author only reported the ablation studies of ConvNet features in Sec. 4.4. Therefore the importance of VQVAE features is less obvious. A naive way to demonstrate its importance is to build a baseline model with ConvNet features only and without autoregressive modeling by directly regressing the action. Comparisons to such a baseline model should be included.

2. Lack of analysis of the inference latency. As a larger model size comes at the cost of higher inference latency, the inference speed of both small and medium models should be included.

3. Lack of comparison of scaling properties with other methods. As shown in Table 1, the results of MILE and the proposed method are comparable with 32 hours of training data. However, it's unclear how the MILE performs with a higher volume of data (such as 256h).  The authors are encouraged to draw the performance curve of both MILE and GAP for comparison, with different scales of data.

**Questions:**

Please see the weakness above.

---

### Meta-Review · Area_Chair_zyqZ · 2024-12-18

**Metareview:**

The paper proposes a novel architecture, GAP, aiming to merge scene generation with planning for scalable driving performance, with an emphasis on reducing the reliance on extensive perception annotations. This is showcased through improved performance with larger model and data sizes, and the introduction of a dual-branch image encoder is highlighted as a key contribution.

The strengths of this work include its innovative integration of scene generation with planning, which shows potential for scalability and reduction in annotation requirements. Furthermore, the paper is well-presented, with comprehensive visualizations that facilitate understanding.

However, there are notable weaknesses. The paper lacks critical ablation studies and an in-depth analysis of the importance of VQVAE features. Additionally, there is an inadequate evaluation of tokenization effects and the application of the method to real-world datasets. Concerns were also raised about the absence of comprehensive latency reporting, which is essential for assessing the feasibility of real-time applications. Finally, the omission of leaderboard evaluation results and some technical inaccuracies detract from the paper's impact.

The decision to reject is mainly due to these unaddressed concerns, particularly the insufficient evaluation on critical components like VQVAE features and real-world dataset applicability, which impact the paper's contributions to the field.

**Additional Comments On Reviewer Discussion:**

During the discussion, Reviewer PHnB highlighted the absence of ablation studies and latency analysis. Reviewer ka22 pointed out the overstatement of the self-supervision claim and questioned the impact of ConvNet features on the results. Reviewers AoXu and GDwn emphasized the need for a more comprehensive evaluation and comparison with other methods.

In response, the authors presented additional experiments to showcase the significance of combining VQVAE features with next-token prediction, which addressed PHnB's concerns to some extent. Although they provided latency results, this only partially addressed the issue. The authors recognized the necessity for action annotations, which clarifies the self-supervision aspect, but this only partially assuages ka22's concerns. The inability to compare scaling properties with methods like MILE, due to dataset constraints, remains a point of contention for AoXu and GDwn.

In reaching the final decision, while PHnB's concerns about ablation studies were somewhat resolved, the partial addressment of latency issues, the unresolved comprehensive evaluation, and the overstated self-supervision claim ultimately led to skepticism about the paper's readiness for acceptance.

---

### Decision · Program_Chairs · 2025-01-22

Reject